# Hyperbolic Representation Learning for Spatial Biology: Capturing Cell Type Hierarchies in Breast Cancer

**Youssef Wally[1], Johan Mylius-Kroken[1], Michael Kampffmeyer[1,2],**

**Rezvan Ehsani[3], Vladan Milosevic[3], Elisabeth Wetzer[1]**

[1]Department of Physics and Technology, UiT The Arctic University of Norway, Tromsø, Norway
[2]Norwegian Computing Center, Oslo, Norway
[3]Department of Clinical Medicine, University of Bergen, Bergen, Norway
{youssef.m.wally,johan.m.kroken,elisabeth.wetzer,michael.c.kampffmeyer}@uit.no
{v.milosevic,rezvan.ehsani}@uib.no

## Abstract

Hyperbolic representation learning has recently emerged as a powerful framework for modeling hierarchical structures in data, often outperforming Euclidean embeddings. We investigate its utility for analyzing high-dimensional biological data from Imaging Mass Cytometry (IMC) of breast cancer tissues. We embed cells into Euclidean and Lorentzian latent spaces via a fully hyperbolic variational autoencoder (VAE) and propose an information-theoretic framework based on k-nearest neighbor estimators to quantify clustering quality using mutual information (MI) and conditional mutual information (CMI). Results show that Lorentzian embeddings preserve substantially more biologically relevant structure compared to Euclidean ones. We further provide open-source tools for Lorentzian MI estimation and hyperbolic UMAP visualization, enabling geometry-aware representation learning for spatial biology. Code available at: https://github.com/youssefwally/FlatlandandBeyond

## 1 Introduction

Encoding multiscale and hierarchical structure has long been a central goal in representation learning. Hyperbolic representation learning models, which operate in negatively curved spaces, have recently been shown to naturally capture hierarchies and outperform Euclidean models across diverse domains ranging from natural language and knowledge graphs to computer vision and recommender systems [1–3]. By embedding data in spaces that mirror tree-like relationships, hyperbolic methods achieve better clustering and classification performance for inherently hierarchical data [4, 5].

These properties make hyperbolic geometry especially appealing for biological applications, where cell types and states often follow complex hierarchical organization. Multiplexed imaging techniques such as Imaging Mass Cytometry (IMC) capture dozens of protein markers at subcellular resolution, enabling high-dimensional single-cell profiling [6]. Accurately modeling the relationships between cell types and functional states in such data requires representations that respect this underlying hierarchy, an area where hyperbolic embeddings hold significant promise [7].

However, despite growing interest, rigorous quantitative comparisons between Euclidean and hyperbolic embeddings remain scarce. Most studies rely on qualitative visualization rather than quantitative geometry-agnostic measures. To address this gap, we propose an information-theoretic evaluation framework that quantifies clustering quality across geometric spaces using mutual information (MI)

39th Conference on Neural Information Processing Systems (NeurIPS 2025) Workshop: MedEurIPS 2025.

and conditional mutual information (CMI) utilizing the Kraskov–Stögbauer–Grassberge (KSG) MI estimator [8]. Applying this framework to a 42-marker breast cancer IMC dataset, we show that Lorentzian (hyperbolic) embeddings capture substantially more biologically relevant structure than Euclidean ones. We further release open-source tools extending the KSG MI estimator [8] to Lorentzian manifolds and enabling UMAP [9] visualizations with hyperbolic distance metrics.

## 2 Methodology

Traditional quantitative metrics such as the Silhouette Score or Average Distortion Index [10, 11] assume Euclidean geometry; linear distances, convex neighborhoods, and isotropy. These assumptions fail in hyperbolic spaces, where distances grow exponentially, and local curvature which affects neighborhood structure. Even substituting Euclidean distances with geodesics can yield misleading results due to the indefinite nature of the Lorentzian inner product and curvature-dependent spread.

Similarly, visualization methods like t-SNE and UMAP [12, 9] exhibit bias towards Euclidean geometry. Thus, assessing which geometry better captures biologically meaningful structure requires evaluation methods that do not assume Euclidean geometry.

We adopt a non-parametric MI estimator based on $k$-nearest neighbor (kNN), specifically the KSG estimator [8], which can operate on arbitrary metric spaces, including Lorentzian geodesics.

**Geometry-Agnostic:** MI and CMI can be estimated directly from pairwise distances, independent of curvature, convexity, or coordinate representation [13]. This allows fair comparison between embeddings learned in Euclidean and hyperbolic spaces.

**Local and Density-Aware:** Unlike global clustering scores, kNN-based MI captures local density variations and neighborhood consistency.

**Cross-Geometry Alignment:** By estimating $I(X;Y)$ (MI), where $X$ and $Y$ denote Euclidean and hyperbolic representations respectively, we quantify the shared information between representations, providing a direct measure of structural preservation.

### 2.1 KSG Estimator Formulation

Given random variables $X$, $Y$, and $Z$, the CMI under the KSG estimator can be expressed as

$$I(X;Y|Z) \approx \psi(k) + \psi(N) - \frac{1}{N} \sum_{i=1}^{N} \left[ \psi(n_x^{(i)} + 1) + \psi(n_y^{(i)} + 1) - \psi(n_z^{(i)} + 1) \right]$$

where $\psi(\cdot)$ is the digamma function, and $n_x^{(i)}, n_y^{(i)}, n_z^{(i)}$ denote the number of neighbors within the $\varepsilon_i$-ball of the corresponding variables, excluding the query point. The radius $\varepsilon_i$ is defined as the maximum distance to the $k$-th nearest neighbor in the joint space. The mutual information (MI) case follows directly by omitting the (Z)-dependent term.

## 3 Data

We use the Imaging Mass Cytometry (IMC dataset from [7], featuring a 42-marker panel for phenotypic and spatial profiling of the tumor microenvironment, with emphasis on cancer-associated fibroblasts in breast cancer. Hierarchical cell annotations span four levels; we use the first three, from broad categories (Cancer, Immune, Endothelial, Fibroblasts) to fine-grained immune subtypes and detailed T cell and macrophage phenotypes. We also benchmark our method on the MNIST handwritten digit dataset [14], a standard test bed for representation learning and clustering, offering a controlled setting to validate geometry-aware embeddings.

## 4 Experiments

### 4.1 Implementation Details

Experiments were conducted in PyTorch [15] using Riemannian optimization [16] via Geoopt [17], with 32-bit precision as in [18]. Models include Hyperbolic Variational Autoencoder (HVAE), and

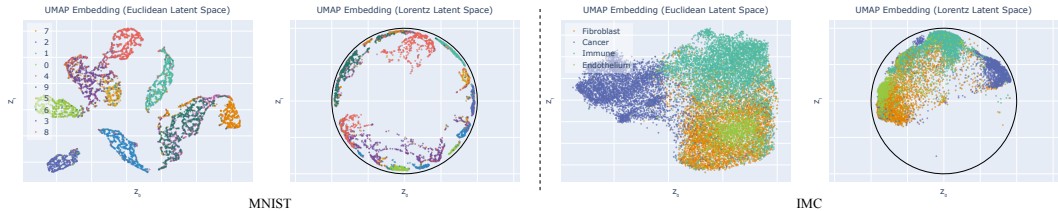

Figure 1: 2D latent embeddings from VAEs in Euclidean and Lorentzian spaces, colored by ground-truth labels.

Table 1: Estimated MI and CMI on IMC and MNIST test sets. Higher is better.

| Quantity | IMC | MNIST |
|---|---|---|
| $MI(D_L;C)$ | **1.07** | **1.86** |
| $MI(D_E;C)$ | 0.96 | 1.78 |
| $MI(D_L;D_E)$ | 0.01 | 4.03 |
| $CMI(D_L;C \mid D_E)$ | **1.06** | **0.16** |
| $CMI(D_E;C \mid D_L)$ | 0.00 | 0.09 |

Euclidean Variational Autoencoder (EVAE). **All analyses are performed on the test set.** To ensure fair comparison, H-VAE and E-VAE are trained independently, with reconstruction loss as a common objective.

# 5 Results

## 5.1 Qualitative Analysis

Visualizations of Euclidean and Lorentzian embeddings (Fig. 1) reveal clear structural differences that highlight the representational advantages of hyperbolic geometry. In Lorentzian space, clusters appear more compact and hierarchically organized, consistent with the space's exponential volume growth. In the IMC dataset, minority classes such as Endothelial Cells (8.40% of total samples) form tighter, more separable clusters than in Euclidean space. This indicates that Lorentzian embeddings capture fine-grained biological distinctions even among underrepresented cell types.

Similar behavior is observed in MNIST, where ambiguous digits such as certain "3"s are positioned between clusters of visually similar digits ("0", "6", "8"), reflecting Lorentz space's ability to represent semantic uncertainty. In contrast, Euclidean embeddings enforce flatter separations that obscure such relationships.

## 5.2 Quantitative Analysis

We evaluate how well each geometry encodes class-relevant structure using MI between pairwise distance matrices Lorentzian Distances ($D_L$), Euclidean Distances ($D_E$) and class labels ($C$). We also compute CMI to quantify the incremental information each geometry contributes beyond the other.

The MI results confirm that Lorentzian embeddings encode more class-relevant information ($MI(D_L;C) > MI(D_E;C)$) in both datasets. The near-zero $MI(D_L;D_E)$ on IMC indicates that the two geometries capture largely non-overlapping structural information. Moreover, $CMI(D_L;C \mid D_E) = 1.06$ versus $CMI(D_E;C \mid D_L) = 0.00$ shows that Lorentzian geometry provides additional, non-redundant information beyond what Euclidean structure explains, demonstrating superior expressiveness and alignment with biological hierarchies.

**Additional studies and statistical analyses (kNN evaluations, McNemar test) are provided in the Supplementary Material. Code available at:** `https://github.com/youssefwally/FlatlandandBeyond`

# 6 Potential Negative Societal Impact

This work uses de-identified biological imaging data to evaluate geometric representation learning. While the methods proposed are purely analytical, future applications to clinical datasets could raise privacy and ethical concerns if data are not properly anonymized. Moreover, improved clustering and cell-type inference could be misused to predict sensitive biological traits without appropriate oversight. To mitigate these risks, transparency, reproducibility, and responsible data use should be prioritized.

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
