# OpenReview forum: "Hyperbolic Representation Learning for Spatial Biology: Capturing Cell Type Hierarchies in Breast Cancer"
_EurIPS.cc/2025/Workshop/MedEurIPS — EurIPS 2025 Workshop MedEurIPS Submission_

### Official Review · Reviewer_hT3f · 2025-10-26
**Hyperbolic Representation Learning for Spatial Biology: Capturing Cell Type Hierarchies in Breast Cancer**

**Rating:** 6
**Confidence:** 3

**Review:**

This work introduced an information-theoretic framework to evaluate the clustering quality of representation learning methods for biological data from Imaging Mass Cytometry (IMC) of breast cancer tissues. The new evaluation approach gave insights into the comparisons between Euclidean and hyperbolic embeddings, and showed the advantages of the latter.

Pros:
- The motivation is interesting.
- The evaluation framework is novel.

Cons:
- More baseline methods and conventional metrics should be included.
- The method should be analyzed theoretically.

---

### Official Review · Reviewer_rfeJ · 2025-10-31
**Review comments**

**Rating:** 6
**Confidence:** 3

**Review:**

This paper presents a methodological contribution by evaluating Hyperbolic Representation Learning (Lorentzian VAEs) for Imaging Mass Cytometry data.

Strength: The core strength lies in an information-theoretic evaluation framework utilizing the kNN-based KSG Mutual Information estimator. Quantitative result confirms that the hyperbolic space provides unique, non-redundant information.

---

### Decision · Program_Chairs · 2025-10-31

**Decision:**

Accept (Poster)

**Comment:**

Both reviewers appreciate the paper’s originality and relevance, highlighting its novel mutual-information–based evaluation of hyperbolic embeddings and its clear application to spatial biology.